# Prevalence and risk factors for feather-damaging behavior in psittacine birds: Analysis of a Japanese nationwide survey

**Kazumasa Ebisawa**[1,2]* , **Shunya Nakayama**[1‡], **Chungyu Pai**[1‡], **Rie Kinoshita**[1‡], **Hiroshi Koie**[1]

**1** College of Bioresource Sciences, Nihon University, Fujisawa, Kanagawa, Japan, **2** Yokohama Bird Clinic, Yokohama, Kanagawa, Japan

☯ These authors contributed equally to this work.
‡ These authors also contributed equally to this work.
* yokohamabirdclinic1997@gmail.com

**Data Availability Statement:** All relevant data are within the manuscript and its S1–S4 Files.

## Abstract

A case control study was conducted to estimate the prevalence of feather-damaging behavior and evaluate the correlation with risk factors among pet psittacine birds in Japan. Although feather-damaging behavior among pet parrots is frequently observed in Japan, its prevalence and potential risk factors have not been investigated. Therefore, we conducted an online questionnaire survey on parrot owners throughout Japan to examine regional differences in feather-damaging behavior and associated risk factors. In total, 2,331 valid responses were obtained. The prevalence of feather-damaging behavior was 11.7%, in general agreement with prior studies. The highest prevalence was among Cockatoos (*Cacatua* spp., etc.; 30.6%), followed by Lovebirds (*Agapornis* spp.; 24.5%) and African grey parrots (*Psittacus erithacus*; 23.7%). Multivariate logistic regression was carried out to calculate the adjusted odds ratio ($OR_{adj}$) for potential risk factors and adjust the confounding of the variables. The odds of feather-damaging behavior were significantly higher for Conures (*Aratinga* spp., *Pyrrhura* spp., *Thectocercus acuticaudatus*, *Cyanoliseus patagonus*) ($OR_{adj}$ = 2.55, $P$ = 0.005), Pacific parrotlets (*Forpus coelestis*) ($OR_{adj}$ = 3.96, $P$ < 0.001), African grey parrots ($OR_{adj}$ = 6.74, $P$ < 0.001), Lovebirds ($OR_{adj}$ = 6.79, $P$ < 0.001) and Cockatoos ($OR_{adj}$ = 9.46, $P$ < 0.001) than Budgerigars (*Melopsittacus undulatus*), and for young adults ($OR_{adj}$ = 1.81, $P$ = 0.038) and adults ($OR_{adj}$ = 3.17, $P$ < 0.001) than young birds, and for signs of separation anxiety ($OR_{adj}$ = 1.81, $P$ < 0.001). Species, bird age and signs of separation anxiety were significantly higher risk factors for feather-damaging behavior than any other potential risk factors. Our findings, which include broad species diversity, are a good source of data for predicting risk factors for feather-damaging behavior and could be useful in preventing declines in welfare.

**Funding:** The authors received no specific funding for this work.

**Competing interests:** The authors have declared that no competing interests exist.

## Introduction

Feather-damaging behavior (FDB) is a behavioral disorder generally seen in parrots kept captive as pets [1, 2]. FDB is a troublesome problem for pet owners, caregivers, and clinicians, and generally indicates poor welfare [1, 3, 4]. FDB includes picking, plucking, chewing, fraying, and biting [5–7], and may also include self-mutilation of skin or muscles, which can inhibit the normal regrowth of feathers [8]. The prevalence of FDB appears to vary among the more than 200 bird species that are commonly kept in captivity [9, 10]. The prevalence of FDB among parrots has been estimated to be 10–17.5% [11–14].

FDB may be an intentional way of coping with stress due to an unsuitable environment and poor management [8, 15]. The causes of FDB have been reported to have origins in boredom (e.g., deprivation of environmental enrichment or foraging opportunities, unsuitable cage size or design) [9, 16–18], environmental stress (e.g., always caged, living with other parrots) [1, 14, 19], loneliness (e.g., social isolation, absence of the preferred owner) [7, 20, 21], separation anxiety [7, 9, 22] and sexual frustration (e.g., delayed reproductive behavior) [18, 23]. In addition, sex [14, 19], age (adult) [13], acquisition source (rescued/rehomed and pet store) [19, 22], hand-rearing [14, 21], being out of the cage for more than 8 hours [22] and sleeping time (more than 8 hours) [19] have been suggested as risk factors for FDB. Conversely, it has been suggested that interacting with people for more than 4 hours a day may help prevent FDB [22].

Problems regarding the understanding of the mechanisms underlying FDB are related to the relative lack of controlled studies on FDB in pet birds and limited veterinary medical knowledge of feather loss and FDB [24]. More accurate information on FDB could not only facilitate better treatment of affected birds, but also lead to the prevention of FDB onset. The purpose of the study reported here was to estimate the prevalence of FDB and evaluate its correlation with risk factors among pet psittacine birds in Japan. We focused on how the presence of humans, conspecific cage mates, and other birds or animals affects FDB because parrots are highly social animals [9, 25]. We investigated whether separation anxiety can cause FDB and whether the presence of humans, conspecific birds, other birds or animals can prevent FDB. Although FDB among pet parrots is frequently observed in Japan, to our knowledge, its prevalence and potential risk factors have not been investigated.

## Materials and methods

### Study population and data collection

Since the approval of human subjects research is required only for medical research in Japan, the Ethics Review Committee of the College of Bioresource Sciences of Nihon University determined that approval is not required for conducting this study. An online questionnaire was compiled using Google Forms (https://www.google.com/forms/about/). All participants were recruited from across Japan through an advertisement on the authors' website (https://www.yokohamabirdclinic.jp), Internet forums, and social networks. Participants were able to respond to up to five birds if they had more than one bird. The survey was carried out online for 16 weeks, from October 2018 to January 2019. Participants were asked to consent to inform before responding the questionnaire. Consent was obtained by click the Yes button and type the participant's name in the box (S1 File). This study was conducted with great care to ensure the privacy and confidentiality of the participants, and to guarantee that the data would be used only for the purposes of scientific research. To avoid any bias in the numbers of FDB or non-FDB responses, the participants were told that the purpose of the survey was research into problem behavior.

## Questionnaire

An original questionnaire form was developed on the basis of FDB risk factors listed in prior studies [12–14, 19, 22] and review papers [7, 26]. There was a total of 26 questions (Table 1). The bird species, not limited to parrots, was selected from a pull-down list. If the species were not on the list or the exact species was not known, the owner could input the information into the text box manually. The wing clipping was asked to confirm if the bird could fly. Wild-caught birds were not listed as an acquisition source because they are not available in pet stores in Japan. A question item on the frequency of bathing/spraying was included to assess the relationship between the frequency of feather wetting and FDB, with "Rarely" defined as once a month or less, "Weekly" as about once a week, and "Daily" as almost every day. Fresh foods were designated as vegetables and/or fruits. Human foods were designated as foods excluding fresh foods, with "Sometimes" defined as about once a week and "Always" as almost every day. FDB was defined as feather picking, plucking, chewing, or biting. Stereotyped behavior was defined as excessive self-grooming, incessant screaming, wire chewing, sham chewing, beak rubbing, food manipulation, wing flapping, pacing, perch circles, corner flips, or route tracing [7, 27, 28]. Reproductive behavior was defined as courtship behavior, copulation behavior, or nesting [29]. The terms used for each behavior were supplemented by a description (S2 File). Signs of separation anxiety were defined as vocalization and/or locomotor activity (e.g., pacing, wing flapping) when the owner leaves home or decreased appetite and/or destructiveness while the owner is absent [30–32]. The presence or absence of signs of separation anxiety was determined by the owner based on the behavior of the bird when the owner leaves home and the condition inside the cage when the owner returns home.

## Data analyses

In total, 3,392 sets of responses were obtained. Responses corresponding to any of the following were excluded from the analysis: duplicated responses, responses that were defective in any way (e.g., unknown species, sex, age, acquisition source and rearing method), responses for hybrid species, and responses for non-psittacine species (e.g., Sparrows, Pheasants, Pigeons, Owls). After removing responses from owners of unknown age, all stray birds were excluded. In addition, responses showing FDB from the time of the acquisition were excluded because the environment at the time of the onset of FDB was unknown. Some of the congeners or closely related species were grouped together because of their similar prevalence of FDB. Species with a proportion of less than 2% were excluded to clarify the trends in FDB according to species or groups. The overall prevalence of FDB was calculated after this exclusion. Finally, 2,331 sets of responses were included in the analysis, which is provided as a S3 File. The true species before grouping is listed in the S4 File.

The genus *Agapornis* was grouped as Lovebirds. The genera *Aratinga*, *Cyanoliseus*, and *Pyrrhura* were grouped as Conures. The genera *Cacatua*, *Calyptorhynchus*, *Eolophus*, and *Lophochroa* were grouped as Cockatoos. Although Cockatiels are Cacatuidae, they were not grouped into Cockatoos because of their small size and large sample size. As the biological significance of chronological age differs between species, age was classified by species-specific life stages as follows: juvenile/adolescent (period after weaning and fledging, until sexual maturation), young adult (early period of sexual maturation), and adult (period after full sexual maturation) [13]. Some criteria were excluded, and similar levels were pooled to increase the sample sizes. "Not caged" was removed from "Conspecific cage mate" and "Cage covered at night". "Parent-rearing" and "Parent-rearing with neonatal handling" were pooled into "Parent-rearing" for the rearing method. "Never" and "Rarely" were pooled into "Rarely" for Bathing/spraying.

**Table 1. Overview of 26 questions of the questionnaire and predictors for pet parrots in Japan.**

| Questions | Available options | Predictors |
|---|---|---|
| What is your gender? | Male | Gender |
| | Female | |
| What is your age? | Pull-down list | Owner age |
| Are you married? | Yes | Married |
| | No | |
| Do you have any children? | Yes | Children |
| | No | |
| How many people are there in your family? | Pull-down list | Family size |
| What is the species? | Pull-down list or free text box | Species |
| What sex is the bird? | Male | Sex |
| | Female | |
| | Don't know | |
| How old is the bird? | Pull-down list (includes Don't know) | Bird age |
| Are you clipping wing feathers? | Yes | Wing clipping |
| | No | |
| How did you acquire the bird? | Pet store/breeder | Acquisition source |
| | Bred at home | |
| | Adopted | |
| | Stray bird | |
| | Don't know | |
| How was the bird reared? | Hand-rearing by owner | Rearing methods |
| | Hand-rearing by pet store or breeder | |
| | Parent-rearing | |
| | Parent-rearing with neonatal handling | |
| | Don't know | |
| Do you have any other birds and/or animals? | Yes | Other birds and/or animals |
| | No | |
| Is there any conspecific cage mate? | Yes | Conspecific cage mate |
| | No | |
| | Not caged | |
| How many hours a day do you let the bird outside of the cage and interact with it? | Pull-down list or Does not let outside | Time let outside of the cage and interaction |
| How many hours do you let the bird sleep at night? | Pull-down list | Sleeping time |
| Do you cover the entire cage at night? | Yes | Cage covered at night |
| | No | |
| | Not caged | |
| How many hours are there with no human presence in a typical day? | Pull-down list | Time with no human presence |
| Do you use ultraviolet light? | Yes | Ultraviolet light |
| | No | |
| How often do you bathe and/or spray the bird? | Never | Bathing/spraying |
| | Rarely | |
| | Weekly | |
| | Daily | |

(*Continued*)

**Table 1.** (Continued)

| Questions | Available options | Predictors |
|---|---|---|
| What are the staple foods? | Only seeds | Staple foods |
| | Only pellets | |
| | Seeds & pellets | |
| Do you feed the bird any fresh foods? | Yes | Fresh foods |
| | No | |
| How often do you feed the bird human foods? | Always | Human foods |
| | Sometimes | |
| | Never | |
| Is feather-damaging behavior observed? | Yes | Feather-damaging behavior |
| | No | |
| Is stereotyped behavior observed? | Yes | Stereotyped behavior |
| | No | |
| Has reproductive behavior been observed within the past 6 months? | Yes | Reproductive behavior |
| | No | |
| Are signs of separation anxiety observed? | Yes | Separation anxiety |
| | No | |

"Sometimes" and "Always" were pooled into "Yes", and "Never" was changed to "No" for Human foods.

Univariate logistic regression was first used to compare potential risk factors between FDB and non-FDB parrots, estimate odds ratios (ORs), and calculate 95% confidence intervals (CIs). The phi coefficient or Cramer's V was used to measure correlations between categorical variables where $P < 0.05$. A predictor was selected based on biological plausibility, the significance of associations with FDB and model fit, when the correlation between variables was very strong (phi or Cramer's V > 0.25) [33]. Multivariate logistic regression was used to calculate the adjusted ORs ($OR_{adj}$) and 95% CIs for potential risk factors for FDB and to adjust the confounding of the variables. The forward selection method based on the likelihood ratio was used to select the variables. Predicted probabilities, Predicted group membership, Standardised residuals and Cook's were checked on save option. Classification plots, Hosmer–Lemeshow goodness of fit and CI for exp(B) were checked on options. Variables were left in the final model when $P < 0.05$ [34]. All nonsignificant variables were tested in the final model to check for residual confounding [35]. Regression diagnostics were performed on the full adjusted analyses using the Hosmer–Lemeshow test for goodness of fit [36]. All statistical analyses were performed using SPSS for Windows (version 20.0; SPSS Inc., Chicago, IL, USA).

## Results

The overall prevalence of FDB was 11.7% (272/2,331 responses) and varied cross-species and groups. The highest FDB prevalence was 30.6% for Cockatoos, followed by 24.5% for African grey parrots and 23.7% for Lovebirds. The prevalence of FDB for Budgerigars, which received the highest number of responses, was 4.9%, followed by Cockatiels, at 7.6% (Table 2).

Ten of the 26 predictors in univariate analysis were significantly associated with increased or decreased ORs for FDB (Tables 3 and 4). Some correlations were found between variables (Table 5). The P values of the Chi-square test derived for each correlation were also shown in Table 5. In the correlation between the presence of children and family size (Cramer's V = 0.591, $P < 0.001$), family size was selected based on model fit. In the correlation between

**Table 2. Sample sizes included in the studied population (n = 2,331) and prevalence of feather-damaging behavior (FDB) by parrot species and group.**

| Species | Sample size | | Prevalence of FDB | | |
|---|---|---|---|---|---|
| | n | % | n | % | 95% CI |
| Budgerigars (*Melopsittacus undulatus*) | 853 | 36.6 | 42 | 4.9 | 3.6–6.6 |
| Cockatiels (*Nymphicus hollandicus*) | 608 | 26.1 | 46 | 7.6 | 5.6–10.0 |
| Lovebirds (*Agapornis* spp.)[a] | 470 | 20.2 | 115 | 24.5 | 20.6–28.6 |
| Conures (various species)[b] | 126 | 5.4 | 14 | 11.1 | 6.2–17.9 |
| Pacific parrotlets (*Forpus coelestis*) | 79 | 3.4 | 14 | 17.7 | 10.0–27.9 |
| Cockatoos (various species)[c] | 72 | 3.1 | 22 | 30.6 | 20.2–42.5 |
| Barred parakeets (*Bolborhynchus lineola*) | 64 | 2.7 | 5 | 7.8 | 2.6–17.3 |
| African grey parrots (*Psittacus erithacus*)[d] | 59 | 2.5 | 14 | 23.7 | 13.6–36.6 |

[a]*Agapornis roseicollis, A. personata, A. fischeri.*

[b]*Aratinga* spp., *Cyanoliseus* sp., *Pyrrhura* spp.

[c]*Cacatua* spp., *Calyptorhynchus* sp., *Eolophus* sp., *Lophochroa* sp.

[d] Include *Psittacus erithacus timneh.*

CI = confidence interval.

species and bathing/spraying (Cramer's V = 0.316, $P < 0.001$), species was selected based on the significance of the association with FDB. In the correlation between bird age and acquisition source (Cramer's V = 0.515, $P < 0.001$), bird age was selected based on biological plausibility.

**Table 3. Results of univariate analysis of owner characteristics associated with feather-damaging behavior (FDB) in 2,331 parrots.**

| Variable | FDB | Non-FDB | Prevalence of FDB | OR | 95% CI | *P*-value |
|---|---|---|---|---|---|---|
| | n | n | % | | | |
| **Gender** | | | | | | |
| Male | 19 | 173 | 9.9 | | | |
| Female | 253 | 1,886 | 11.8 | 1.22 | 0.75–2.00 | 0.425 |
| **Owner age** | | | | | | |
| 18–39 years | 50 | 408 | 10.9 | | | |
| 40–49 years | 133 | 902 | 12.9 | 1.20 | 0.85–1.70 | 0.294 |
| ≥ 50 years | 89 | 749 | 10.6 | 0.97 | 0.67–1.40 | 0.970 |
| **Married** | | | | | | |
| No | 103 | 802 | 11.4 | | | |
| Yes | 169 | 1,257 | 11.9 | 1.05 | 0.81–1.36 | 0.730 |
| **Children** | | | | | | |
| No | 197 | 1,32173 | 13.0 | | | |
| Yes | 75 | 8 | 9.2 | 0.68 | 0.52–0.90 | 0.007* |
| **Family size** | | | | | | |
| One | 42 | 331 | 11.3 | | | |
| Two | 128 | 773 | 14.2 | 1.30 | 0.90–1.89 | 0.160 |
| Three | 66 | 487 | 11.9 | 1.07 | 0.71–1.61 | 0.754 |
| Four or more | 36 | 468 | 7.1 | 0.61 | 0.38–0.97 | 0.036* |

*$P < 0.05$.

OR = odds ratio.

CI = confidence interval.

**Table 4. Results of univariate analysis of bird characteristics associated with feather-damaging behavior (FDB) in 2,331 parrots.**

| Variable | FDB | Non-FDB | Prevalence of FDB | OR | 95% CI | P-value |
|---|---|---|---|---|---|---|
| | n | n | % | | | |
| **Species** | | | | | | |
| Budgerigars (*Melopsittacus undulatus*) | 42 | 811 | 4.9 | | | |
| Cockatiels (*Nymphicus hollandicus*) | 46 | 562 | 7.6 | 1.58 | 1.03–2.43 | 0.038* |
| Barred parakeets (*Bolborhynchus lineola*) | 5 | 59 | 7.8 | 1.64 | 0.62–4.29 | 0.317 |
| Conures (various species)[a] | 14 | 112 | 11.1 | 2.41 | 1.28–4.56 | 0.007* |
| Pacific parrotlets (*Forpus coelestis*) | 14 | 65 | 17.7 | 4.16 | 2.16–8.01 | < 0.001* |
| African grey parrots (*Psittacus erithacus*) | 14 | 45 | 23.7 | 6.01 | 3.06–11.80 | < 0.001* |
| Lovebirds (*Agapornis* spp.)[b] | 115 | 355 | 24.5 | 6.26 | 4.30–9.10 | < 0.001* |
| Cockatoos (various species)[c] | 22 | 50 | 30.6 | 8.50 | 4.71–15.30 | < 0.001* |
| **Bird sex** | | | | | | |
| Male | 143 | 1,240 | 10.3 | | | |
| Female | 129 | 819 | 13.6 | 1.37 | 1.06–1.76 | 0.016* |
| **Bird age** | | | | | | |
| Juvenile/adolescent[d] | 18 | 227 | 7.3 | | | |
| Young adult[e] | 90 | 866 | 9.4 | 1.31 | 0.77–2.22 | 0.314 |
| Adult[f] | 164 | 966 | 14.5 | 2.14 | 1.29–3.56 | 0.003* |
| **Wing clipping** | | | | | | |
| No | 248 | 1,915 | 11.5 | | | |
| Yes | 24 | 144 | 14.3 | 1.29 | 0.82–2.02 | 0.274 |
| **Acquisition source** | | | | | | |
| Pet store/breeder | 217 | 1,754 | 11.0 | | | |
| Bred at home | 34 | 177 | 16.1 | 1.55 | 1.05–2.30 | 0.028* |
| Adopted | 21 | 128 | 14.1 | 1.33 | 0.82–2.15 | 0.252 |
| **Rearing methods** | | | | | | |
| Hand-rearing by owners | 159 | 1,166 | 12.0 | | | |
| Hand-rearing by pet store or breeder | 100 | 799 | 11.1 | 0.92 | 0.70–1.20 | 0.527 |
| Parent-rearing (including neonatal handling) | 13 | 94 | 12.1 | 1.01 | 0.56–1.85 | 0.963 |
| **Lives with other birds and/or animals** | | | | | | |
| No | 81 | 600 | 8.3 | | | |
| Yes | 191 | 1,459 | 11.6 | 0.97 | 0.74–1.28 | 0.828 |
| **Conspecific cage mate** | | | | | | |
| No | 247 | 1,840 | 11.8 | | | |
| Yes | 25 | 219 | 9.8 | 0.85 | 0.55–1.31 | 0.465 |
| **Time let outside of the cage and interaction (h)** | | | | | | |
| < 2 | 175 | 1,271 | 12.1 | | | |
| ≥ 2 | 86 | 693 | 11.0 | 0.90 | 0.69–1.19 | 0.458 |
| Not let outside | 11 | 95 | 10.4 | 0.84 | 0.44–1.60 | 0.598 |
| **Sleeping time (h)** | | | | | | |
| < 8 | 52 | 468 | 10.0 | | | |
| 8–12 | 158 | 1,172 | 11.9 | 1.21 | 0.87–1.69 | 0.252 |
| > 12 | 62 | 419 | 12.9 | 1.33 | 0.90–1.97 | 0.151 |
| **Cage covered at night** | | | | | | |
| No | 48 | 358 | 11.8 | | | |
| Yes | 224 | 1,701 | 11.6 | 0.98 | 0.71–1.37 | 0.915 |
| **Time with no human presence (h)** | | | | | | |
| < 3 | 76 | 508 | 13.0 | | | |

*(Continued)*

**Table 4.** (Continued)

| Variable | FDB | Non-FDB | Prevalence of FDB | OR | 95% CI | *P*-value |
|---|---|---|---|---|---|---|
| | n | n | % | | | |
| 3–7 | 74 | 665 | 10.0 | 0.74 | 0.53–1.05 | 0.088 |
| 7–11 | 79 | 651 | 10.8 | 0.81 | 0.58–1.13 | 0.222 |
| > 11 | 43 | 235 | 15.5 | 1.22 | 0.82–1.83 | 0.329 |
| **Ultraviolet light** | | | | | | |
| No | 231 | 1,756 | 11.6 | | | |
| Yes | 41 | 303 | 11.9 | 1.03 | 0.72–1.47 | 0.876 |
| **Bathing/spraying** | | | | | | |
| Rarely | 71 | 864 | 7.6 | | | |
| Weekly | 173 | 1,043 | 14.2 | 2.02 | 1.51–2.70 | < 0.001* |
| Daily | 28 | 152 | 15.6 | 2.24 | 1.40–3.59 | < 0.001* |
| **Staple foods** | | | | | | |
| Only seeds | 53 | 563 | 8.7 | | | |
| Only pellets | 56 | 322 | 14.8 | 1.85 | 1.24–2.76 | 0.003* |
| Seeds and pellets | 163 | 1,174 | 12.2 | 1.47 | 1.06–2.04 | 0.019* |
| **Fresh foods** | | | | | | |
| No | 52 | 401 | 11.5 | | | |
| Yes | 220 | 1,658 | 11.7 | 1.02 | 0.74–1.41 | 0.889 |
| **Human foods** | | | | | | |
| No | 206 | 1,702 | 10.8 | | | |
| Yes | 66 | 357 | 16.6 | 1.53 | 1.13–2.06 | 0.006* |
| **Reproductive behavior** | | | | | | |
| No | 85 | 647 | 11.6 | | | |
| Yes | 187 | 1,412 | 11.7 | 1.01 | 0.77–1.32 | 0.954 |
| **Stereotyped behavior** | | | | | | |
| No | 224 | 1,717 | 11.5 | | | |
| Yes | 48 | 342 | 12.3 | 1.06 | 0.77–1.50 | 0.667 |
| **Separation anxiety** | | | | | | |
| No | 160 | 1,411 | 10.2 | | | |
| Yes | 112 | 648 | 14.7 | 1.52 | 1.18–1.97 | < 0.001* |

[a]*Aratinga* spp., *Cyanoliseus* sp., *Pyrrhura* spp.

[b]*Agapornis roseicollis*, *A. personata*, *A. fischeri*.

[c]*Cacatua* spp., *Calyptorhynchus* sp., *Eolophus* sp., *Lophochroa* sp.

[d]Small parrot: < 5 months, medium parrot: < 1 year 11 months, large parrot: < 4 years 11 months.

[e]Small parrot: 5 months to 3 years 11 months, medium parrot: 2 years to 5 years 11 months, large parrot: 5 years to 10 years 11 months.

[f]Small parrot: > 4 years, medium parrot: > 6 years, large parrot: > 11 years.

*$P < 0.05$.

OR = odds ratio.

CI = confidence interval.

Seven predictors—family size, species, bird age, bird sex, staple food, human foods, and separation anxiety—were included in the multivariable model. The final model included 3 predictors (species, bird age, and separation anxiety) in the multivariate logistic regression. Compared with Budgerigars, the odds of FDB were 2.5 times higher in Conures ($OR_{adj} = 2.55$, $P = 0.005$), almost four times higher in Pacific parrotlets ($OR_{adj} = 3.96$, $P < 0.001$), almost seven times higher in African grey parrots ($OR_{adj} = 6.74$, $P < 0.001$), almost seven times higher in Lovebirds ($OR_{adj} = 6.79$, $P < 0.001$), and 9.5 times higher in Cockatoos ($OR_{adj} = 9.46$,

**Table 5. Phi and Cramer's V coefficient for correlations between variables where $P < 0.05$ in the univariate analysis.**

| | Presence of children | Family size | Species | Bird sex | Bird age | Acquisition Source | Bathing/spraying | Staple foods | Human foods |
|---|---|---|---|---|---|---|---|---|---|
| Family size | 0.591[a] | | | | | | | | |
| | 0.000** | | | | | | | | |
| Species | 0.105 | 0.107 | | | | | | | |
| | 0.001* | 0.000** | | | | | | | |
| Bird sex | −0.140† | 0.037 | 0.090 | | | | | | |
| | 0.449 | 0.354 | 0.009* | | | | | | |
| Bird age | 0.090 | 0.070 | 0.209 | 0.014 | | | | | |
| | 0.000** | 0.001* | 0.000** | 0.788 | | | | | |
| Acquisition source | 0.067 | 0.043 | 0.123 | 0.019 | 0.515[a] | | | | |
| | 0.005* | 0.184 | 0.000** | 0.654 | 0.000** | | | | |
| Bathing/spraying | 0.070 | 0.062 | 0.316[a] | 0.098 | 0.075 | 0.034 | | | |
| | 0.004* | 0.006* | 0.000** | 0.000** | 0.000** | 0.254 | | | |
| Staple foods | 0.037 | 0.066 | 0.249 | 0.022 | 0.077 | 0.015 | 0.089 | | |
| | 0.210* | 0.002* | 0.000** | 0.564 | 0.000** | 0.910 | 0.000** | | |
| Human foods | 0.001† | 0.039 | 0.230 | 0.050† | 0.029 | 0.076 | 0.040 | 0.066 | |
| | 0.953 | .316 | 0.000** | 0.016 | 0.383 | 0.001* | 0.157 | 0.006* | |
| Separation anxiety | 0.029† | 0.032 | 0.212 | 0.000† | 0.076 | 0.097 | 0.036 | 0.069 | 0.081 |
| | 0.166 | 0.488 | 0.000** | 0.994 | 0.001* | 0.000** | 0.220 | 0.004* | 0.000** |

Above the cell shows the Phi and Cramer's V coefficient,

†Phi,

[a]Cramer's V > 0.25.

Below the cell shows the P values for the Chi-square test,

*$P < 0.01$,

**$P < 0.001$.

$P < 0.001$). Compared with juveniles/adolescents, the odds of FDB were almost two times higher in young adults ($OR_{adj} = 1.81$, $P = 0.038$) and almost three times higher in adults ($OR_{adj} = 3.17$, $P < 0.001$). Signs of separation anxiety ($OR_{adj} = 1.81$, $P < 0.001$) were significantly correlated with FDB (Table 6).

## Discussion

### Owner characteristics

The greatest response to the present survey was from older women, which is in line with prior studies [12, 22, 37]. There is a possibility of a potential response bias because older women respond more cooperatively to Internet surveys, or are just very skilled at keeping animals and enjoy it as primary caregivers. We hypothesized that if birds perceived a human family as a flock, then the characteristics of the owner would be related to FDB. The presence of children and family size were likely to decrease the odds of FDB; however, this trend did not remain in the final model. Further studies of owner characteristics such as the personality of the owner, type of human-bird interaction and the purpose of keeping a bird may further clarify the impact on FDB.

### Regional differences in FDB prevalence and trends in species

The prevalence of FDB among parrots in Japan was estimated at 11.7%, and this largely agrees with prior studies on captive parrots [12–14]. The majority of responses were for small-size

**Table 6. Final model of the multivariate analysis of risk factors significantly associated with feather-damaging behavior in 2,331 parrots.**

| Variable | B | SE | OR$_{adj}$ | 95% CI | *P* value |
|---|---|---|---|---|---|
| **Species** | | | | | |
| Budgerigars (*Melopsittacus undulatus*) | | | | | |
| Cockatiels (*Nymphicus hollandicus*) | 0.402 | 0.225 | 1.49 | 0.96–2.32 | 0.074 |
| Barred parakeets (*Bolborhynchus lineola*) | 0.457 | 0.498 | 1.58 | 0.60–4.19 | 0.358 |
| Conures (various species)[a] | 0.936 | 0.332 | 2.55 | 1.33–4.89 | 0.005* |
| Pacific parrotlets (*Forpus coelestis*) | 1.377 | 0.339 | 3.96 | 2.04–7.71 | < 0.001* |
| African grey parrots (*Psittacus erithacus*) | 1.908 | 0.360 | 6.74 | 3.33–13.65 | < 0.001* |
| Lovebirds (*Agapornis* spp.)[b] | 1.916 | 0.195 | 6.79 | 4.64–9.95 | < 0.001* |
| Cockatoos (various species)[c] | 2.247 | 0.321 | 9.46 | 5.05–17.73 | < 0.001* |
| **Bird age** | | | | | |
| Juvenile/adolescent | | | | | |
| Young adult | 0.591 | 0.286 | 1.81 | 1.03–3.16 | 0.038* |
| Adult | 1.154 | 0.275 | 3.17 | 1.85–5.44 | < 0.001* |
| **Separation anxiety** | | | | | |
| No | | | | | |
| Yes | 0.595 | 0.144 | 1.81 | 1.37–2.40 | < 0.001* |

[a]*Aratinga* spp., *Cyanoliseus* sp., *Pyrrhura* spp.

[b]*Agapornis roseicollis*, *A. personata*, *A. fischeri*.

[c]*Cacatua* spp., *Calyptorhynchus* sp., *Eolophus* sp., *Lophochroa* sp.

*P < 0.05.

B = partial regression coefficient.

SE = standard error.

OR$_{ajd}$ = adjusted odds ratio.

CI = confidence interval.

parrots (e.g., Budgerigars, Cockatiels, Lovebirds), and only a small proportion of responses was seen for medium-size and large parrots. Previous studies conducted in the US, UK, and Italy have shown a high proportion of medium- (e.g., Senegal parrots [*Poicephalus senegalus*], Quaker parrots [*Myiopsitta monachus*]) and large-size parrots (e.g., African grey parrots, Cockatoos, Amazon parrots [*Amazona* spp.], Macaws [*Ara* spp.]) [12–14]. It is of considerable interest that even though the species and numbers of individuals included in the survey population of the present study in Japan were largely different from those of prior studies, despite different captive environments and management in different countries, no large difference was seen in the prevalence of FDB. In addition, species-specific and group FDB prevalence also generally agreed with those in previous studies (Table 2) [12–14].

Species was shown to be a significant risk factor in the final model. There were significant differences among species due to interactions among several variables. Species in diverse other taxa are known to differ in their relative vulnerabilities to abnormal repetitive behaviors, including FDB [38], and our findings were consistent with this assumption. Species were also a significant risk factor in other studies, although there were differences in the variables analyzed simultaneously [13]. Our findings also suggest that a certain number of FDB may be observed in each species, regardless of differences in environment and management among countries.

## Bird age

Age was found to be a significant risk factor for FDB. The findings suggest that increasing age may be a risk factor for FDB, which is in agreement with the results of prior studies [13]. It has been suggested that most captive parrots show behavioral problems as soon as they reach sexual maturity, and it has been proposed that sexual maturity is the key phase of the onset of FDB in parrots [39]. It has also been proposed that FDB becomes increasingly prevalent in young adulthood, with the incidence plateauing as birds enter adulthood [13]. The higher odds seen in this study for FDB among adults ($OR_{adj}$ = 3.17, $P$ < 0.001) compared with young adults ($OR_{adj}$ = 1.81, $P$ = 0.038) is consistent with these proposals. However, young adults were not a significant variable in univariate analysis (OR = 1.31, $P$ = 0.314), though were a significant variable in the final model. The finding suggests that age is not the only risk factor for FDB, but other factors may play a role. A follow-up cohort study may be needed to clarify further the interrelationships between risk factors of FDB and age-related changes, as the current environment may have changed since the onset of FDB.

## Sleeping and cage covered at night times

Sleeping time is potentially very important for FDB. The perception that insufficient sleeping time may contribute to FDB is widely accepted [40]. Conversely, sleeping more than 8 hours has been reported to increase the odds for FDB significantly in African grey parrots [19]. In this study, neither sleeping nor cage covered at night time affected the odds for FDB. However, sleeping time was not clearly defined in this study; thus, the owners may have answered the question for sleeping time based on time the cage was covered, assuming that the bird was resting despite still being in a noisy environment. To investigate the association between sleep time and FDB more accurately, it is recommended to define sleeping time as starting from the time the bird is placed in a dark and uninterrupted quiet area for sleep, and to inquire about the environmental conditions when the cage is covered.

## Reproductive behavior

Although sexual frustration has been suggested as a cause of FDB [18, 23], no correlation with FDB was observed in this study. However, the owners may have overlooked reproductive behavior or been unable to identify reproductive behaviors accurately in their birds. Some reproductive behaviors may be simply displacement behaviors that have no correlation with reproductive activity. For example, tearing up paper and substrate has been shown to be related to a lack of foraging opportunities [16]. Therefore, to investigate the relationship between reproductive behavior and FDB more accurately, comparing sex hormone levels in the droppings and/or blood of birds with and without FDB may be necessary.

## Rearing method

No difference was seen in the prevalence of FDB by rearing method. It has been reported that FDB is more common in hand- than in parent-reared birds [14, 21]. Alternatively, the number of parrots that were parent-reared may not have been numerous enough to detect any significant differences compared with hand-reared parrots. The proportion of parent-reared parrots with FDB in this study, including those that were handled during rearing, was very small (n = 13). Rearing method is a vital question involving the welfare of birds, and is therefore essential for investigating differences in FDB onset between rearing methods and determining the optimal rearing method for each species.

## Signs of separation anxiety

Signs of separation anxiety were a significantly higher risk factor for FDB. Gaskins and Hungerford [22] suggested that adoption (i.e., rescued or rehomed) may be a risk factor for separation anxiety, which could be the underlying cause of FDB. However, in this study, adoption showed no significant difference in the odds of FDB between acquisition sources. We defined signs of separation anxiety by observing behavior when the owner left home or was absent because we predicted that daily rather than permanent separation was the cause of FDB. It has been suggested that loneliness is a cause of FDB [7, 20, 21] because parrots are highly social and depend on flocks in the wild [9, 25]. In this study, neither family size, time with no human presence, presence of other birds and/or animals, nor presence of conspecific cage mates affected the odds of FDB. If birds prefer humans, separation anxiety may occur even if conspecifics are present. In addition, we did not collect information about relationships with other birds or animals. If the birds were afraid of other birds or potential predators such as dogs or cats, this could be a potential risk factor, and the presence of other birds or animals would not prevent separation anxiety. This result also involves interaction with other variables, therefore, to investigate further the environment that causes separation anxiety, it will be necessary to ask whether birds prefer humans or conspecifics and to clarify the relationships with other birds or animals.

## Limitations of the study

A survey carried out over the Internet is likely to involve a response bias, so the reliability of the survey results are potentially limited [41]. A limitation of the present study is that rather than a definitive diagnosis of FDB being given by a veterinarian, FDB was diagnosed by the owner. FDB status may have included medical issues (e.g., infection, parasites, skin diseases, neoplasms), reproductive behavior (e.g., brood patch), and molting. It is also possible that cases of latent feather picking or slight feather fraying not clear to the owners were overlooked, or that the owner did not know the correct species. Similar-looking species, such as the genera *Amazona* and *Pyrrhura*, or hybrids, may have led to erroneous responses by the owners. Furthermore, it may be more difficult even for an experienced bird owner to recognize different kinds of behavior such as stereotyped behavior, reproductive behavior, or signs of separation anxiety and provide reliable data, than for them to provide reliable data for the other parameters. These issues would weaken the power to detect risk factors or lead to the detection of erroneous risk factors.

Although explanations were given for the definitions of the options for question items regarding bathing/spraying and human foods, the owner might have responded to these items subjectively, which would not preserve the reliability of the data. Future surveys should take care to design questions that can be responded to objectively.

It was not clear whether the parrot's current environment and management were the same as that before the onset of FDB. When owners attempted to improve the environment and management for FDB treatment, the survey could not be taken as a risk factor for the onset of FDB. Removing this uncertainty by carrying out a chronological study would provide better data on the relationship between risk factors and FDB. In addition, FDB in parrots is generally regarded as a multifactorial disease that may be influenced by a number of medical, genetic, neurobiological and/or socio-environmental factors [7]. The etiological routes of this study approach are mainly environmental and partly neurobiological; thus, a wide range of other etiological routes needs to investigate as potential causes of FDB.

A cross-species approach provides a good source of information for research into general risk factors for FDB in parrots; however, various species or parameters and the limited number

of species limit the reliability of the data. In addition, multivariate testing or biases between predictors may omit other predictors, so species-specific potential risk factors may not be accurately represented. A species-specific approach is required to reveal more specific information. Clarifying the environment and management at the time of onset of FDB in a chronological study would be of immense clinical value and would likely reveal which species are better suited to captivity, how they should be reared, and what sort of environment and management should be provided. This would likely be valuable information for the improved welfare of captive birds.

## Supporting information

**S1 File. Informed consent form.**
(PDF)

**S2 File. Descriptions of the terms used for each behavior.**
(PDF)

**S3 File. 2,331 sets of responses included in this study.**
(XLSX)

**S4 File. 2,331 sets of responses listed true species.**
(XLSX)

## Acknowledgments

We wish to thank Akitsugu Konno and Yuko Ikkatai for their support in carrying out this study.

## Author Contributions

**Conceptualization:** Kazumasa Ebisawa, Hiroshi Koie.

**Data curation:** Kazumasa Ebisawa, Hiroshi Koie.

**Formal analysis:** Kazumasa Ebisawa, Hiroshi Koie.

**Investigation:** Kazumasa Ebisawa, Shunya Nakayama, Chungyu Pai, Rie Kinoshita, Hiroshi Koie.

**Methodology:** Kazumasa Ebisawa, Shunya Nakayama, Chungyu Pai, Hiroshi Koie.

**Project administration:** Shunya Nakayama, Chungyu Pai, Hiroshi Koie.

**Resources:** Shunya Nakayama, Chungyu Pai, Rie Kinoshita, Hiroshi Koie.

**Software:** Shunya Nakayama, Chungyu Pai, Rie Kinoshita.

**Supervision:** Hiroshi Koie.

**Validation:** Shunya Nakayama, Chungyu Pai, Rie Kinoshita.

**Visualization:** Chungyu Pai, Rie Kinoshita.

**Writing – original draft:** Kazumasa Ebisawa.

**Writing – review & editing:** Kazumasa Ebisawa, Hiroshi Koie.

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
