## [Decision Letter · Decision Letter 0]

19 Mar 2021

PONE-D-21-02618

Prevalence and risk factors for feather-damaging behavior in psittacine birds: Analysis of a Japanese nationwide survey

PLOS ONE

Dear Dr. Ebisawa,

Thank you for submitting your manuscript to PLOS ONE. After careful consideration, we feel that it has merit but does not fully meet PLOS ONE’s publication criteria as it currently stands. Therefore, we invite you to submit a revised version of the manuscript that addresses the points raised during the review process.

Please see the reviewer comments below for specific changes to be made during the revision process.

We look forward to receiving your revised manuscript.

Kind regards,

I Anna S Olsson, Ph.D.

Academic Editor

PLOS ONE

Journal Requirements:

'NO - The funders had no role in study design, data collection and analysis, decision to publish, or preparation of the manuscript.'

Reviewers' comments:

Reviewer's Responses to Questions

**Comments to the Author**

1. Is the manuscript technically sound, and do the data support the conclusions?

Reviewer #1: Yes

Reviewer #2: Yes

2. Has the statistical analysis been performed appropriately and rigorously? 

Reviewer #1: Yes

Reviewer #2: Yes

3. Have the authors made all data underlying the findings in their manuscript fully available?

Reviewer #1: No

Reviewer #2: Yes

4. Is the manuscript presented in an intelligible fashion and written in standard English?

Reviewer #1: Yes

Reviewer #2: Yes

5. Review Comments to the Author

Reviewer #1: This paper is well-written, and based on a very nice dataset that is also well-analysed.

Please see attached document for our complete comments on this manuscript. At this stage I recomend 'Major Revision'

Reviewer #2: As the authors highlighted, feather-damaging behaviour is a serious welfare concern in pet psittacine birds, with still a poor understanding of the risk factors to this behavioural problem; therefore the study has merit to help further our understanding of the potential causes of this problem, and the authors acknowledge the limitations of the study and approach used.

The manuscript is generally well written, clear and to the point. There are a few sentences that need to be checked for English grammar.

Specific comments:

L36: I think you should add the basis for comparison, at least for “young adults” and “adults”, so add “compared to young birds”, as right now we don’t know what those 2 categories compare to.

L37-38: “significantly highly” does not make sense, maybe “significantly higher” but the authors still need to state the basis for comparison, so higher than which other factors, or state that those factors were significant risk factors.

L68-69: not all parrots species live in stable flocks in the wild, some live in pairs, other associates with others at various times of the year depending on, for instance, food availability. Hence, this statement could be nuanced in that parrots are social animals.

L107-108: it could be useful to add the definition provided for those terms, in the text or as a supplementary data file.

Tables 3 and 4: I would find it useful to add a column to show the prevalence of FDB within each species or variable presented; right now you show % across the FDB and non-FDB parrots but this still makes it difficult to compare the prevalence within each species/variable (so comparing the FDB to non-FDB as % within each species) . Adding a column to show this per variable reported would help.

Table 5: I think you have a typo with twice “Bird sex” in the first row, and the second one should probably be corrected for “Bird age”.

L201-202 and elsewhere: I think you should replace “Separation anxiety” by “Signs of separation anxiety”, since separation anxiety per se can probably not be confidently assessed by the owners, but signs of it could be recognized. It is also unclear how separation anxiety was assessed: you mention L295 that it was “by observing behaviour when the owner left home or was absent”, but on which basis were these behavioural observations then conducted? Video-camera recording?

L222-223: I think you could also add the type of human-bird interaction also as an important owner characteristic, given that the human-animal relationship has been shown to have an important impact on the welfare of other animal species, and may have more direct influences on the bird than the owner’s personality for instance.

L.274-275 and elsewhere: “In addition, regurgitation may not be unique to courtship, as it also occurs in anxious or sick birds”: a few statements in the discussion appear to be unsubstantiated such as this one. Please add a reference to support this statement, or some other facts, or alternatively remove those.

L335: should it be “cross-species”?

6. PLOS authors have the option to publish the peer review history of their article (what does this mean?). If published, this will include your full peer review and any attached files.

Reviewer #1: **Yes: **Georgia Mason

Reviewer #2: **Yes: **Jean-Loup Rault

---

## [Author Response · Author response to Decision Letter 0]

12 Apr 2021

I Anna S Olsson, Ph.D.

Academic Editor

PLOS ONE

April 09, 2021

Dear Dr. I Anna S Olsson:

Thank you for inviting us to submit a revised draft of our manuscript entitled, “Prevalence and Risk Factors for Feather-damaging Behavior in Psittacine Birds: Analysis of a Japanese Nationwide Survey” to Journal of PLOS ONE. We also appreciate the time and effort you and each of the reviewers and editor have dedicated to providing insightful feedback on ways to improve our paper. Thus, it is with great pleasure that we resubmit our article for further consideration. We have incorporated changes that reflect the detailed suggestions that you graciously provided. We also hope that our changes and responses below satisfactorily address the issues and concerns that you raised.

To facilitate your review of our revisions, the following is a point-by-point response to the comments delivered in your letter dated March 20, 2021.

Response to reviewer #1

Thank you for your review of our paper. We have answered each of your points below.

METHODS

1. Did you need human ethics approval, for surveying owners? (In my country one would need that)

Response: Approval of human subjects research is required only for medical research in Japan. We confirmed with the Ethics Review Committee of the College of Bioresource Sciences of Nihon University whether approval was necessary for this study, they replied that it was not necessary. We mentioned this in lines 80-83.

2. Re identifying species, any idea how accurate owners were? Was doubt here a reason why “Some of the congeners or closely related species were grouped together because of their similar prevalence of FDB”.? (Lines 121 – 123). 

Response: You have raised an important point; We had no way of ascertaining the accuracy of the species response. However, In Japan, the owners usually know the exact species of their birds because pet stores tell owners the exact species. Therefore, we did not doubt the response.

3. Also then in Tables 4 - 6, the term “species” does not really mean “species” then? If not it should be changed to some else (e.g. “Species/genus”??)

Response: Thank you for this suggestion. We revised “species” to “Species/genus” on Tables 4 and 6.

4. Lines 122- 124: “Species with a proportion of less than 2% were excluded to clarify the trends in FDB according to species or groups”. I understand this is because you want to ID the risk factors for FDB, but can you clarify whether your overall prevalence value of 11.7% was calculated before or after this exclusion?

Response: Thank you for this suggestion. We mentioned in line 130 that the overall prevalence was calculated after this exclusion.

5. Lines 146-147: “The forward selection method based on the likelihood ratio was used to select the variables”.: can you give more details so that someone else could replicate what you did? 

Response: Thank you for this suggestion. We added SPSS option settings in lines 153-155.

RESULTS

6. Table 5: The interactions need explaining: it’s great that you ran them, but it's not enough to just present the P values – please let the reader know what the pattern was that lead to each significant interaction. 

Response: We agree with you and added “The P values of the Chi-square test derived for each potential interaction were also shown in Table 5.” to lines 176-177.

7. Also is one of the two “bird sex” column really “bird age”?

Response: We agree with you and added “type of human-bird interaction” to lines 238-239.

8. Furthermore, where an interaction between terms A and B is significant, that means it’s not OK to talk about any significant main effects of terms A or B, because they probably are just driven by the interaction. This has considerable interactions for the Discussion. 

Response: We agree with you and removed “and acquisition source showed only a weak interaction with signs of separation anxiety (Cramer’s V = 0.097, P < 0.001)” and “We cautiously propose that signs of separation anxiety may be affected by species. In fact, signs of separation anxiety showed a strong interaction with species (Cramer’s V = 0.212, P < 0.001)”.

DISCUSSION

9. This is interesting and well written, but needs work because it presents effects of “species” (really “species and genus”), “age” and other effects are presented as though they are simple main effects, when really they have interactive effects so such an interpretation is not valid. For example, species interacts with 8 other terms (see Table 5), which means that effect of species varies in magnitude with these 8 factors (it may hold for some levels of these but not for other). Likwise bird age interacts with three or more other factors (hard to tell because of a possible typo in the columns names of Table 5), such that there is no general pattern of bird age – it varies according to all these other factors. 

Response: We agree with you and revised those as follows.

Regional differences in FDB prevalence and trends in species

We added “There were significant differences among species due to interactions among several variables” to lines 254-255 and “Species were also a significant risk factor in other studies, although there were differences in the variables analyzed simultaneously [13]” to lines 257-259.

Bird age

We removed “However, in the present study, there was insufficient information to interpret age as a risk factor”.

We added “However, young adults were not a significant variable in univariate analysis (OR = 1.31, P = 0.314), though were a significant variable in the final model. The finding suggests that age is not the only risk factor for FDB, but other factors may play a role. A follow-up cohort study may be needed to clarify further the interrelationships between risk factors of FDB and age-related changes, as the current environment may have changed since the onset of FDB” to lines 270-275.

Signs of separation anxiety

We removed “and acquisition source showed only a weak interaction with signs of separation anxiety (Cramer’s V = 0.097, P < 0.001)” and “We cautiously propose that signs of separation anxiety may be affected by species. In fact, signs of separation anxiety showed a strong interaction with species (Cramer’s V = 0.212, P < 0.001)”.

We added “This result also involves interaction with other variables, therefore,” to lines 322-323.

10. Line 217 – “for companionship as a substitute for children": isn’t this rather sexist? Maybe older women are just very skilled at keeping animals an enjoy it? 

Response: We agree with you and removed “It has been suggested that older women may be overrepresented because they tend to acquire birds for companionship as a substitute for children [38]”. We revised lines 233-235 to “There is a possibility of a potential response bias because older women respond more cooperatively to Internet surveys, or are just very skilled at keeping animals and enjoy it as primary caregivers.”

11. Line 294 “ acquisition source showed only a weak interaction with separation anxiety (Cramer’s V = 0.097, P < 0.001).” Isn’t a P that small indicative of a rather strong interaction? Also what does this interaction mean i.e. what causes it? (Again more information needs to be provided in the Results).

Response: We agree with you revised those as above (Signs of separation anxiety).

Response to reviewer #2

Thank you for your comments. Our answers to your points are as follows.

1. L36: I think you should add the basis for comparison, at least for “young adults” and “adults”, so add “compared to young birds”, as right now we don’t know what those 2 categories compare to.

L37-38: “significantly highly” does not make sense, maybe “significantly higher” but the authors still need to state the basis for comparison, so higher than which other factors, or state that those factors were significant risk factors.

Response: Thank you for this suggestion. We revised “significantly highly” to “significantly higher”. And we added a comparison not only to bird age, but also to species.

2. L68-69: not all parrots species live in stable flocks in the wild, some live in pairs, other associates with others at various times of the year depending on, for instance, food availability. Hence, this statement could be nuanced in that parrots are social animals.

Response: We agree with you and removed [that live in stable flocks in the wild] from line 70.

3. L107-108: it could be useful to add the definition provided for those terms, in the text or as a supplementary data file.

Response: Thank you for this suggestion. We added a supplementary data file with a descriptions of the terms used for each behavior (S2_File).

4. Tables 3 and 4: I would find it useful to add a column to show the prevalence of FDB within each species or variable presented; right now you show % across the FDB and non-FDB parrots but this still makes it difficult to compare the prevalence within each species/variable (so comparing the FDB to non-FDB as % within each species). Adding a column to show this per variable reported would help.

Response: Thank you for this suggestion. We added a column to show the prevalence of FDB within each species or variable presented and removed the columns of % across the FDB and non-FDB parrots in Tables 3 and 4.

5. Table 5: I think you have a typo with twice “Bird sex” in the first row, and the second one should probably be corrected for “Bird age”.

Response: Thank you for finding the typo. We revised “Bird sex” to “Bird age” in a row heading of Table 5.

6. L201-202 and elsewhere: I think you should replace “Separation anxiety” by “Signs of separation anxiety”, since separation anxiety per se can probably not be confidently assessed by the owners, but signs of it could be recognized. It is also unclear how separation anxiety was assessed: you mention L295 that it was “by observing behaviour when the owner left home or was absent”, but on which basis were these behavioural observations then conducted? Video-camera recording?

Response: We agree with you and have incorporated this suggestion throughout our paper. We mentioned in lines 113-116 when the owner observed the behavior of the bird and conditions inside the cage and the owner did not use the video camera recording.

7. L222-223: I think you could also add the type of human-bird interaction also as an important owner characteristic, given that the human-animal relationship has been shown to have an important impact on the welfare of other animal species, and may have more direct influences on the bird than the owner’s personality for instance.

Response: We agree with you and added “type of human-bird interaction” to lines 238-239.

8. L.274-275 and elsewhere: “In addition, regurgitation may not be unique to courtship, as it also occurs in anxious or sick birds”: a few statements in the discussion appear to be unsubstantiated such as this one. Please add a reference to support this statement, or some other facts, or alternatively remove those.

Response: Thank you for this suggestion. We remove this sentence from line 295.

9. L335: should it be “cross-species”?

Response: We agree with you and revised “across-species” to “cross-species” in lines 162 and 360.

Again, thank you for giving us the opportunity to strengthen our manuscript based on your valuable comments and queries. We have worked hard to incorporate your feedback and hope that you find these revisions sufficient for accepting our submission.

Sincerely,

Kazumasa Ebisawa

Nihon University, College of Bioresource Sciences, Laboratory of Veterinary Physiology

1866 Kameino, Fujisawa, Kanagawa 252-0880, Japan

TEL: +81-466-84-3800

E-mail: yokohamabirdclinic1997@gmail.com

---

## [Decision Letter · Decision Letter 1]

9 May 2021

PONE-D-21-02618R1

Prevalence and risk factors for feather-damaging behavior in psittacine birds: Analysis of a Japanese nationwide survey

PLOS ONE

Dear Dr. Ebisawa,

Thank you for submitting your manuscript to PLOS ONE. After careful consideration, we feel that it has merit but does not fully meet PLOS ONE’s publication criteria as it currently stands. Therefore, we invite you to submit a revised version of the manuscript that addresses the points raised during the review process.

The remaining issues are minor, although important. Reviewer 1 has an important comment about the statistical analysis, which you need to address. As the editor, I find that the text is sometimes a little misleading, which I ask you to address. You will find our detailed comments below.

We look forward to receiving your revised manuscript.

Kind regards,

I Anna S Olsson, Ph.D.

Academic Editor

PLOS ONE

Journal Requirements:

Additional Editor Comments (if provided):

Line 41 "higher risk factors for feather-damaging behavior than which other potential risk factors" is not grammatically correct - do you mean "higher risk factors for feather-damaging behavior than any other potential risk factors"?

Line 54 This sentence begins with "Although the cause of FDB is believed to be psychological stress" and then you present a number of potential origins, of which many are indeed situations of psychological stress. "Although" is misleading in this context, as it suggests that the examples are of situations other than psychological stress. You can simply start the sentence with "The cause of FDB".

Lines 73-75  This paragraph is overall about the aim of your study and what you collected data on. It is misleading to say that you focused on regional differences, as the comparison between countries is not a part of your study, but instead refers to a discussion of your results with those of other studies. Please remove this sentence.

Lines 106-116 and 313-4: The way you refer to separation anxiety gives the impression that you asked the owner to observe the behaviour of the bird specifically in the situation of leaving home and returning home. Did you do this? Or did you simply ask the owner how the bird behaves when they leave home and when they return? If you did ask them specifically to observe behaviour to collect data, then your description is correct. If you asked them how the bird behaves but did not ask them to observe this in a specific way, you need to revise the wording.

Section Limitations of the study

Please reflect on the limitations of owner-reported data for behaviour. How reliably do you expect that owners can distinguish the different kinds of behaviour? I would argue that it is more difficult even for an experienced bird owner to recognise different kinds of behaviour and hence provide reliable data, than for them to provide reliable data for the other parameters you collected information on, so I would expect that there is more noise in the behaviour data.

Reviewers' comments:

Reviewer's Responses to Questions

**Comments to the Author**

1. If the authors have adequately addressed your comments raised in a previous round of review and you feel that this manuscript is now acceptable for publication, you may indicate that here to bypass the “Comments to the Author” section, enter your conflict of interest statement in the “Confidential to Editor” section, and submit your "Accept" recommendation.

Reviewer #1: (No Response)

Reviewer #2: All comments have been addressed

2. Is the manuscript technically sound, and do the data support the conclusions?

Reviewer #1: Partly

Reviewer #2: Yes

3. Has the statistical analysis been performed appropriately and rigorously? 

Reviewer #1: No

Reviewer #2: Yes

4. Have the authors made all data underlying the findings in their manuscript fully available?

Reviewer #1: No

Reviewer #2: Yes

5. Is the manuscript presented in an intelligible fashion and written in standard English?

Reviewer #1: Yes

Reviewer #2: Yes

6. Review Comments to the Author

Reviewer #1: The paper is much improved. The one outstanding issue is that interactions are still handled incorrectly. When a model reveals a significant interaction, ADDITIONAL ANALYSES need to be run to investigate what drives it. An interaction means one of three things: an apparent effect is stronger in some sub-groups than others; an apparent effect is only present in some sub-groups; or an apparent effect flips direction in some sub-groups compared to others. The only way to find out what's happening is id to drill down into the data and run additional analyses using subsets (sorry).

Also could I ask that when you supply the survey results as an SOM, you upload your values as they were BEFORE you pooled them into your (somewhat arbitrary) taxon groups? That way interested researchers could then see values broken up by true species (not genus or higher), which would be very useful for people interested in running meta-analyses.

Reviewer #2: Thank you for this revised version of the manuscript. You have addressed all my comments and suggestions, and I think that it made the manuscript clearer.

7. PLOS authors have the option to publish the peer review history of their article (what does this mean?). If published, this will include your full peer review and any attached files.

Reviewer #1: **Yes: **Georgia Mason

Reviewer #2: **Yes: **Jean-Loup Rault

---

## [Author Response · Author response to Decision Letter 1]

13 May 2021

I Anna S Olsson, Ph.D.

Academic Editor

PLOS ONE

May 13, 2021

Dear Dr. I Anna S Olsson:

Thank you for inviting us to submit a revised draft of our manuscript entitled, “Prevalence and Risk Factors for Feather-damaging Behavior in Psittacine Birds: Analysis of a Japanese Nationwide Survey” to Journal of PLOS ONE. We also appreciate the time and effort you and each of the reviewers and editor have dedicated to providing insightful feedback on ways to improve our paper. Thus, it is with great pleasure that we resubmit our article for further consideration. We have incorporated changes that reflect the detailed suggestions that you graciously provided. We also hope that our changes and responses below satisfactorily address the issues and concerns that you raised.

To facilitate your review of our revisions, the following is a point-by-point response to the comments delivered in your letter dated May 9, 2021.

Response to Additional Editor

Thank you for your review of our paper. We have answered each of your points below.

1. Line 41 "higher risk factors for feather-damaging behavior than which other potential risk factors" is not grammatically correct - do you mean "higher risk factors for feather-damaging behavior than any other potential risk factors"?

Response: Thank you for correcting the grammar. We revised "higher risk factors for feather-damaging behavior than any other potential risk factors".

2. Line 54 This sentence begins with "Although the cause of FDB is believed to be psychological stress" and then you present a number of potential origins, of which many are indeed situations of psychological stress. "Although" is misleading in this context, as it suggests that the examples are of situations other than psychological stress. You can simply start the sentence with "The cause of FDB".

Response: Thank you for this suggestion. We revised to start the sentence with "The cause of FDB". We remove reference No.18. We renumbered the reference list.

3. Lines 73-75 This paragraph is overall about the aim of your study and what you collected data on. It is misleading to say that you focused on regional differences, as the comparison between countries is not a part of your study, but instead refers to a discussion of your results with those of other studies. Please remove this sentence.

Response: Thank you for this suggestion. We remove this sentence. Reference No.12–14 are cited elsewhere.

4. Lines 106-116 and 313-4: The way you refer to separation anxiety gives the impression that you asked the owner to observe the behaviour of the bird specifically in the situation of leaving home and returning home. Did you do this? Or did you simply ask the owner how the bird behaves when they leave home and when they return? If you did ask them specifically to observe behaviour to collect data, then your description is correct. If you asked them how the bird behaves but did not ask them to observe this in a specific way, you need to revise the wording.

Response: Thank you for this suggestion. We asked the owner how the bird behaves when they leave home and when they return. We revised this sentence to “The presence or absence of signs of separation anxiety was determined by the owner based on the behavior of the bird when the owner leaves home and the condition inside the cage when the owner returns home”.

Section Limitations of the study

5. Please reflect on the limitations of owner-reported data for behaviour. How reliably do you expect that owners can distinguish the different kinds of behaviour? I would argue that it is more difficult even for an experienced bird owner to recognise different kinds of behaviour and hence provide reliable data, than for them to provide reliable data for the other parameters you collected information on, so I would expect that there is more noise in the behaviour data.

Response: Thank you for this suggestion. We added the sentence “it may be more difficult even for an experienced bird owner to recognize different kinds of behavior such as stereotyped behavior, reproductive behavior, or signs of separation anxiety and provide reliable data, than for them to provide reliable data for the other parameters”.

Response to reviewer #1

Thank you for your review of our paper. We have answered each of your points below.

1. The paper is much improved. The one outstanding issue is that interactions are still handled incorrectly. When a model reveals a significant interaction, ADDITIONAL ANALYSES need to be run to investigate what drives it. An interaction means one of three things: an apparent effect is stronger in some sub-groups than others; an apparent effect is only present in some sub-groups; or an apparent effect flips direction in some sub-groups compared to others. The only way to find out what's happening is id to drill down into the data and run additional analyses using subsets (sorry).

Response: I apologize for using the inappropriate term "interaction". We used the phi coefficient or Cramer’s V to measure correlations between categorical variables where P < 0.05. Therefore, we revised the term "interaction" to "correlation". Since multivariate logistic regression was used to adjust the confounding of the variables, we think that sub-group analysis is not necessary. We apologize again for the confusion.

2. Also could I ask that when you supply the survey results as an SOM, you upload your values as they were BEFORE you pooled them into your (somewhat arbitrary) taxon groups? That way interested researchers could then see values broken up by true species (not genus or higher), which would be very useful for people interested in running meta-analyses.

Response: Thank you for this suggestion. We added a supplementary file with the true species and scientific names. We added the sentence “The true species before grouping is described in the S4 file” to Lines 127-128.

Response to reviewer #2

Thank you for your review of our paper.

Reference list

The reference number 18 has been retracted because line 54 has been deleted. Therefore, the reference list has changed. We reviewed reference list to ensure that it is complete and correct.

Again, thank you for giving us the opportunity to strengthen our manuscript based on your valuable comments and queries. We have worked hard to incorporate your feedback and hope that you find these revisions sufficient for accepting our submission.

Sincerely,

Kazumasa Ebisawa

Nihon University, College of Bioresource Sciences, Laboratory of Veterinary Physiology

1866 Kameino, Fujisawa, Kanagawa 252-0880, Japan

TEL: +81-466-84-3800

E-mail: yokohamabirdclinic1997@gmail.com

---

## [Editor Report · Decision Letter 2]

30 Jun 2021

Prevalence and risk factors for feather-damaging behavior in psittacine birds: Analysis of a Japanese nationwide survey

PONE-D-21-02618R2

Dear Dr. Ebisawa,

We’re pleased to inform you that your manuscript has been judged scientifically suitable for publication and will be formally accepted for publication once it meets all outstanding technical requirements.

Kind regards,

I Anna S Olsson, Ph.D.

Academic Editor

PLOS ONE
---

## [Editor Report · Acceptance letter]

2 Jul 2021

PONE-D-21-02618R2 

Prevalence and risk factors for feather-damaging behavior in psittacine birds: Analysis of a Japanese nationwide survey 

Dear Dr. Ebisawa:

I'm pleased to inform you that your manuscript has been deemed suitable for publication in PLOS ONE. Congratulations! Your manuscript is now with our production department. 

Kind regards, 

on behalf of

Dr. I Anna S Olsson 

Academic Editor

PLOS ONE